# Effect of Heat Treatment on Tensile Properties and Microstructure of Co-Free, Low Ni-10 Mo-1.2 Ti Maraging Steel

**DOI:** 10.3390/ma15062136

**Published:** 2022-03-14

**Authors:** Hossam Halfa, Asiful H. Seikh, Mahmoud S. Soliman

**Affiliations:** 1Steel Technology Department, Central Metallurgical R&D Institute (CMRDI), Helwan 11731, Egypt; dr.hossam@cmrdi.sci.eg; 2Department of Physics, College of Science, Shaqra University, Shaqra 15556, Saudi Arabia; 3CEREM, Deanship of Scientific Research, King Saud University, P.O. Box 800, Riyadh 11421, Saudi Arabia; aseikh@ksu.edu.sa; 4Department of Mechanical Engineering, College of Engineering, King Saud University, P.O. Box 800, Riyadh 11421, Saudi Arabia

**Keywords:** electroslag remelting, maraging steel, Co-free, Mo-containing, mechanical properties

## Abstract

Production of high-quality maraging steel is dependent not only on the production technology but also on the alloying design and heat treatment. In this work, cobalt-free, low nickel, molybdenum-containing maraging steel was produced by melting the raw materials in a vacuum induction melting furnace and then refining with a shielding gas electroslag remelting unit. The critical transformation temperatures of the investigated steel samples were determined experimentally by differential scanning calorimetry (DSC) analysis and theoretically aiding Thermo-Calc software. Types and chemical composition plus volume fraction and starting precipitation temperature of suggested constituents calculated with the aid of Thermo-Calc software. The microstructures of forged steel specimens that were heat-treated under several conditions were evaluated by X-ray diffraction (XRD), optical microscopy (OP), scanning electron microscopy (SEM), and electron backscattering (EBSD), in addition to transmission electron microscopy (TEM). The mechanical properties of the investigated steel specimens were evaluated by measuring the tensile strength properties and micro-hardness, furthermore, estimating their fracture surface using scanning electron microscopy at lower magnification. The metallographic results show that the microstructure of steel in aged conditions includes high-alloyed martensite and nickel-rich phase, in addition to the low-alloyed-retained-austenite, intermetallic compounds, and lavas-phase (MoCr). Furthermore, TEM and EBSD studies emphasized that the produced steel has high dislocation density with nano-sized precipitate with an average size of ~19 ± 1 nm. Moreover, the metallographic results show that the mentioned microstructure enhances the tensile properties by precipitation strengthening and the TRIP phenomenon. The tensile strength results show that the n-value of investigated steel passes two stages and is comparable with the n-value of TRIP-steel. Steel characterized by 2100 MPa ultimate tensile strength and uniform elongation of more than 7% can be produced by the investigated production routine and optimum heat treatment conditions.

## 1. Introduction

Maraging steel possesses high ultimate tensile strength and low creep under stress. This steel contains low carbon content. Hardening occurs as the martensite phase transformation arises [1] after being processed with heat treatment. Due to its high mechanical properties, this alloy has a wide application in mold casting, aerospace industries, and tool die materials. After solution treatment and the aging process in the range of 500 to 550 °C, the precipitation of intermetallic particles of fine Ni_3_(Al, Ti, Mo) and Fe_2_Mo occurs [2,3]. Strengthening occurs due to the p recipitation of the intermetallic compound particles. Thus, it exhibits higher toughness compared with the conventional high-carbon martensite alloys. According to recent research, the combined presence of Ni_3_Ti and Fe_2_Mo [4] or Fe_7_Mo_6_ [5] precipitate strengthens Co-free maraging steels with Ni mass contents of 18 wt.%. Rapid formation of Ni_3_Ti occurs as the fast diffusion of titanium atoms takes place [4,5]. 

Solution treatment (ST) and aging treatment (AT) are used to control the strength property of maraging steel [6]. An alloying element is supposed to enter into the lattice substrate. Thus, the distortion of the lattice increases in the case of solution treatment. This phenomenon promotes the martensite transformation and thus increases the martensite transformation. Using the AT, the toughness and strength of the material increased. Plastic deformation was utilized before applying both the ST and AT heat treatment. Plastic deformation leads to an increase in dislocation density and microstructure refinement that gives more energy for precipitate and transformation into martensite [7]. Although aging at high temperatures leads to reverse γ creation, coarse precipitation counteracts its advantageous effects on low-temperature hardness. On the other hand, earlier investigations found that aging at 500 °C gave advanced toughness by escalating the content of the hold/reverse film in cobalt-containing maraging steel [8,9]. Due to the importance of the properties of maraging steel, research in this area has received more attention in contrast to precipitation and austenite formation [10,11,12]. 

Although a large number of phase transformation investigation studies [13,14,15,16,17,18,19,20,21,22,23,24,25] exist in maraging steels, a restricted number of works have been conducted on the physical modeling of the steel precipitate, structure kinetics [26,27], and austenite reversal kinetics [28]. Utilizing the improvement of differential scanning calorimetry (DSC), it was achievable to map out the phase conversion dynamics underneath an embarrassed temperature program. Therefore, DSC has been extensively used to study phase transformation dynamics in diverse systems [29,30,31,32,33,34,35].

Noteworthy, there are several maraging steel classes with different chemical compositions and heat treatments. In addition, the previous work reported that the production of high-quality maraging steel is dependent on the production technology [5,7,36,37,38], alloying design [13,14,15,18], and heat treatment [6,7,8,9,10,11,12,15,16,17,18,19,20,21,22]. In the recent decade, metallurgists and scientists accomplish a significant effort in inventing new methods like the additive manufacturing method for producing maraging steel with high technological properties. The major disadvantage of this method is the limitation of production capacity and size of the working part [36,37,38].

On the other hand, double vacuum remelting and electroslag remelting processes are the primary production methods [5,6,7,13,14,15,16,17,18], although there has been an intensive effort from the metallurgists and scientists for innovating new techniques for producing maraging steels [36,37,38]. Because double vacuum remelting and electroslag remelting methods are suitable for the production of large-scale and heavy-weight ingots, in addition to the technological properties of produced steel, they meet the requirement of end-users.

In this present work, cobalt-free, low nickel molybdenum-containing maraging steel specimens were heat-treated according to the proper temperature obtained experimentally from differential scanning calorimetry (DSC) analysis and theoretically aiding Thermo-Calc software. Samples were cooled at different rates after solution treatment. The samples were air-cooled (AC) with 10 wt.% Mo, which was designated as M10AC. Based on hardness value, the M10AC samples were chosen for further aging treatment. Five different aging temperatures (400 °C, 450 °C, 500 °C, 550 °C, and 600 °C) with three different holding times (60 min, 120 min, and 240 min) for each temperature were taken for the aging process. Different characterization such as XRD, SEM, hardness, and tensile tests were conducted on these samples. Based on the above tests, the final sample was chosen, and further characterizations, including XRD, fractography, tensile, TEM, and EBSD, were conducted.

## 2. Materials and Methods

### 2.1. Production and Material

The investigated steel was produced utilizing raw materials (steel scrap, ferroalloys, etc.). The analyzed raw materials were melted in a vacuum induction furnace. The molten steel was poured into a steel mold after completely melting and adjusting both the chemical composition and temperature. The produced ingot was used as a consumable electrode in a direct current electroslag remelting (ESR) furnace equipped with a shielding gas unit. Remelting of the consumable electrode in the electroslag remelting unit was completed under calcium fluoride-based slag 5% TiO_2_ + 95% (70% CaF_2_; 15% Al_2_O_3_; 15% CaO). An open-air induction furnace energized at 15-kilowatt power and 450-kilohertz frequency at a temperature of approximately 1500 °C was used for preparing the master slag from CaF_2_, CaCO_3_, and Al_2_O_3_ reagent-grade purchased from MERCKS. A. (Merck Limited, an affiliate of Merck KGaA, Darmstadt, Germany, 11/F, Elite Centre, 22 Hung Tong Road, Kwun Tong, Kowloon, Hong Kong, China) to produce slag similar to the chemical composition of electroslag remelting, ESR-type slag. To ensure the high homogeneity of slag powder, the dried and weighted components of the master slag components (CaF_2_, CaCO_3_, and Al_2_O_3_) were mixed on a rotating mill for 30 min in a plastic container. Subsequently, the slag powder was collected and kept in a 250 g plastic package. Precise material balance was done for producing the investigated cobalt-free, low-nickle high-molybdenum maraging steel. Table 1 shows the chemical composition of the studied maraging steel produced by the ESR process with 10 wt.% Mo.

### 2.2. Thermo-Calc Studies

In this method, based on the thermodynamic calculations and knowing the chemical composition of the tested steel, both the chemical composition and the amount of each constituent were easily indicated. The multi-component system Fe-C-Ni-Mo-Cr-Ti was utilized to calculate the thermodynamic equilibria of the investigated steel composition. A sub-lattice model reported in the literature [39,40,41] described the phases in the database. Table 2 tabulates the different constituents of the investigated steel chemical composition at the end of solidification and computed in the theoretical estimations.

### 2.3. Differential Scanning Calorimetry (DSC) 

Phase transformation temperatures of the present steel were measured utilizing DSC analysis (DSC, Labsys Setaram, Caluire, France), using temperature scans, with the rate of 0.66 °C/s in the Argon atmosphere.

### 2.4. Solution Treatment and Age Hardening

Based on DSC results, the solid solution treatment was carried out at 1150 °C for 2 h, followed by air cooling. Aging treatment was carried out at five different aging temperatures (400 °C, 450 °C, 500 °C, 550 °C, and 600 °C) for three aging times of 60 min, 120 min, and 240 min.

### 2.5. Mechanical Properties 

The tensile tests were conducted according to the ASTM E08 standard at room temperature (~20 °C) until complete fracture occurred. Tensile testing apparatuses use servo hydraulic power controlled by a computer. The INSTRON 8000^®^ tensile testing machine (Instron, Norwood, MA, USA) of ±100 kN load limit was used with tensile testing software, which is dedicated to the machine. An extensometer with a gauge length of 25 mm were used to measure the strain at a strain rate of 10−3/sec.

Micro-hardness analysis of the specimens was estimated along the horizontal flat plane utilizing a Vickers micro-hardness tester. During the test, a load measuring of 100 gm was employed for 20 s.

### 2.6. Microstructural Evaluation 

Polishing of the specimen was conducted with different grades of emery paper (120, 180, 1000, and 2000), followed by cloth polishers. Then, the polished specimens were chemically etched utilizing 10% ammonium per-sulfate, then 10% nitric acid in ethyl alcohol solutions. After etching, the investigated steel sample was studied optically using a Leica optical microscope (Leica, Wetzlar, Germany) for microstructure observations. 

A scanning electron microscope (SEM) Model JEOL JSM-6360 (Jeol, Tokyo, Japan) was used to analyze the surface morphology of the studied maraging steel at different experimental conditions. 

XRD was used to characterize the crystalline structure and phases of the tested steel. A Rigaku Ultima III X-ray diffractometer (Rigaku, Tokyo, Japan) was used to record diffraction patterns from monochromatic Cu- Kα radiation operated at 20 °C and a scan rate of 2°/min. The X-ray machine’s built-in software was used to obtain the crystallographic planes of X-ray diffraction. The volume fraction of austenite (Vγ) was specified using the following Equation (1):Vγ = 1.4 ∗ Iγ/[Iα + 1.4 ∗ Iγ] (1)
where Iγ is the average of the integrated intensity from the plan miller indices, such as the (111)γ and (200)γ planes, and Iα is the integrated intensity from the plan miller indices, including the (110)α planes. It was noteworthy that the value 1.4 was the correction factor in Equation (1), which was determined experimentally by many investigators [42,43].

EBSD Analysis utilized A PANalytical PANN analytical X’Pert Pro MRD goniometer (Malvern Panalytical Ltd., Malvern, UK) with a Cu tube operating at 40,000 V was employed to analyze the specimen’s bulk texture. The steel micro and crystallographic texture structures were explored in the rolling (RD) and transverse direction (TD) sections. Different steps of the investigated steel specimen preparation are reported and published elsewhere [44].

A JEOL JEM 3010 TEM (Jeol, Tokyo, Japan) was utilized, which worked at a 300 kV speeding up voltage. The TEM studied circle with a thickness of 500 μm (0.5 mm) is taken from the cross portion of expelled billets (ED-plane) and precisely ground down to around 15 μm thick for TEM examinations; 3 mm plates were embedded from the specimens and cleaned to a hole utilizing a twin-stream electro-cleaning plant at 25 °C under zero degree and 15 V, using a solution of 30% nitric acid in methanol. Selected area electron diffraction (SAED) patterns were taken from a 2 μm breadth region. The specimen was cut into meager foils for TEM examination. TEM foils were cleaned at −45 °C, 10 V, in 75% (HNO_3_) nitric acid and 25% ethanol preparation, and afterward tested in a Philips CM200 electron magnifying lens with a Gatan 300 W CCD camera.

## 3. Results and Discussion

### 3.1. Thermo-Calc Study 

Figure 1 describes the solidification behavior of the investigated steel. The solidification behavior of the studied steel specimen was calculated theoretically by the thermodynamic-equilibria model. The thermodynamic-equilibria model anticipated the chemical compositions, the volume fraction of different precipitated phases, and the starting precipitation temperature of different phases during the cooling of the studied steel. Data extracted from Figure 1 were collected and tabulated in Table 3. The data extracted from the solidification-path diagrams show that a high molybdenum content in the studied steel encourages precipitation of laves phase-C14 and retained austenite. Table 3 reports that 17.5 wt.% laves phase-C14 was precipitated at a temperature around 1090 °C, while 11 wt.% retained-austenite start precipitation at a temperature of 650 °C. 

The expected chemical compositions of different constituents at room temperature are presented in Figure 2. Figure 2 shows that laves phase-C14 with 56 wt.% Mo and 44 wt.% Cr were precipitated; it also shows that FCC_A2#2 carbide (titanium carbide, TiC) is the only precipitated carbide in this steel chemical composition. The dominant phase was the BCC_A2 (α-martensite), consisting predominantly of Fe and traces of other elements (Figure 2a). The second phase is the austenite (FCC_A1), consisting predominantly of nickel and a minute quantity of additional elements, as shown in Figure 2c. The laves phase (LAVES-C14) consisted mainly of Mo, Fe, Ni, Ti, and Cr. The intermetallic compounds consisted predominantly of Fe, Ti, Ni, Mo, and Si. The principal discriminator was, thus, a high Ti content, together with Si. The intermetallic compounds suggested from utilizing Thermo-Calc software were Ni_3_Ti and (Ni-Mo)_3_Ti.

### 3.2. DSC Analysis

Differential scanning calorimetry, DSC diagram shows the thermic behavior of materials during heating. However, these thermic peaks are dependent on the transformation of the phases. The DSC examination of the present steel specimens was analyzed and is displayed in Figure 3. Five fundamental peaks are visible on the DSC diagram of the studied steel specimen. However, the first exothermic peak typically happened due to the martensite recovery and the retained-austenite evolution [39,40,41]. The second exothermic peak appears with continued heating of the investigated steel, which accelerates the decomposing of the intermetallic compound, such as Ni_3_(Ti, Mo) [45]. Ni_3_(Ti, Mo) are representative in clusters from 400 to 550 °C, according to Tewari et al. [45]. 

The third exothermic peak is typically visible due to phase transformation from martensite to austenite. This transformation occurs at a temperature range from 600 to 700 °C, depending on the molybdenum contents in the studied steel. Furthermore, the last two peaks are compared to the interaction between the structure and the circumstances of the solid solution. In the high temperature, peaks appear due to the dissociation of the precipitate laves phase-C14. The DSC results were interpreted with the aid of Thermo-Calc studies in the prior section, Figure 1 and Table 3. The DSC curve for the studied steel specimen shows a few shifts at high temperatures, which could be induced by the effect of impurities, such as oxides or sulfides. 

The maraging steel was aged (tempered) at a temperature range of 400–600 °C, and the solution heat treatment temperature was >1083 °C, according to the DSC curve. It is noteworthy that the studied steel contains high alloying element contents; thus, microsegregation dominated. Therefore, the studied steel requires high homogenized temperatures. Differential scanning calorimetry results recommended that the solution heat treatment temperature be increased to more than 1083 °C, in this work, to ensure a more successful heat treatment.

The results extracted from the DSC diagrams show that the high molybdenum content in studied steel has a remarkable effect in different transformation temperatures, as shown in Figure 3. The previous results were confirmed by aided Thermo-Calc studies, as in the prior section and Figure 1.

### 3.3. Aging Treatment 

Thermo-calc and DSC studies recommended that the studied steel requires high homogenization temperatures so that the solution heat treatment temperature be increased to 1150 °C in this work to ensure a more successful heat treatment.

Figure 4 shows the hardness variation of the investigated steel with aging time and temperature. The Figure 4 result is explained by an increase in the hardness that may be attributed to the intermetallic compounds strengthening mechanism of the steel sample during aging. The hardness profile of the studied steel samples increased gradually with increasing aging temperature up to the peak hardness, as shown in Figure 4. The peak hardness appeared at 500 °C. Increasing temperature beyond the peak hardness temperature led to a sharp decrease in the hardness values. This decrease in hardness occurs due to the dissociation of the intermetallic and laves-phase, as explained in the Thermo-Calc studies section. Zhang et al. [46] stated that as the aging temperature rises, the volume fraction of blocky retained-austenite in martensite blocks increases. In addition, the retained-austenite phase deteriorates the studied steel toughness. The peak hardness value indicates the optimum requirement of the aging process (temperature and time). From Figure 4, the authors decided the optimum parameters of the heat treatment process to get the designed mechanical properties. Table 4 shows the optimum heat treatment conditions for the production of the studied steel; it is anticipated that tensile and elongation properties will be enhanced as the material ages at optimum conditions.

The solid-solution-annealing (austenitization) temperature of 1150 °C was assumed to be sufficient to prevent the formation of any precipitates at some stage in homogenization annealing or soaking during forging operations. Therefore, all of the lavas-phase would precipitate during the subsequent cooling after finishing the heat treatment (solid solution annealing) or forged-processing. Conversely, the precipitation of the rest of the intermetallic compounds, such as Ni_3_Ti, Ni_3_(Ti-Mo), Ni_3_Mo, etc., should only occur during the aging heat treatment for all three conditions. The thermo-calc studies recommended the mentioned heat treatment temperatures. The forging and homogenization, in addition to the aging temperatures, are summarized in Table 4. Furthermore, the forging and homogenization, in addition to the aging cycle, are schematically represented in Figure 5. 

### 3.4. Microstructural Analysis

Figure 6 shows the optical micrographs of the present steel after ST and in the peak-aged conditions at 500 °C for different times. The microstructure in the peak aged conditions consisted of martensite packets within prior-austenite grains. The transformed-austenite-grains were distinguished because of the preferential etching along their boundaries. Therefore, the martensite-packets within the austenite grain did not extend beyond the respective prior-austenite grain boundary. 

The substructure of the martensite could not be monitored because of the fineness of the martensite laths. During the aging of the steels under investigation, the well-known precipitation reactions lead to hardening. The initial precipitation in cobalt-free molybdenum-containing maraging steel at 480 °C occurs as Ni_3_Mo, which on prolonged aging is replaced by either lava-phase (Fe_2_Mo) or the σ phase (FeMo) [4]. Since the alloy additionally contains titanium as an auxiliary hardener, the precipitation of Ni_3_Ti has also been reported; alternatively, it has been suggested that part of the titanium may be present in the molybdenum precipitate, i.e., as Ni_3_(Mo, Ti). This microstructure suggested that the mechanical properties of the studied steel were enhanced due to the well-known substructure of lathe martensite consisting predominantly of a high density of dislocation tangles within the laths [13,14].

### 3.5. XRD Analysis

Figure 7 shows XRD patterns for the studied heat-treated maraging steel samples. Furthermore, Figure 7 emphasizes the present microstructure consisting of martensite, retained austenite, and different precipitated particles. Figure 7 shows that the highest intensity peak was present in the patterns corresponding to the martensite (body-centered cubic) phase for all the studied steel samples. Figure 7 shows that the produced maraging steel primarily consists of the martensite phase. The martensite formed because of the phase transformation process and alloying elements’ contents. The X-ray diffraction pattern of the austenite peaks in the studied heat-treated steels reflects the high stability of the retained austenite, as shown in Figure 7. The high stability of the retained austenite could be attributed to the highly alloyed element contents. Additional small peaks correspond to the nickel intermetallic compounds and laves-phase, such as Ni_3_Al ((001)–24.92°, (101)–38.54°), Ni_3_Ti ((101)–22.82°), Ni_3_Fe ((001)–25.06°, (101)–38.79°), or Ni_3_Mo ((101)–26.73°, (200)–35.50°), as shown in Figure 7. The exact type of the precipitate particles could not be identified from the XRD patterns since the primary peaks were close together. However, all phase peaks with small intensities in a 2-theta angle ranging between 20° and 40° were expected to represent nickel intermetallic compounds or laves-phase. 

However, these small phase peaks were significantly noticed in the aged sample at 240 min, but the same peaks with lower intensities were also noticeable in all the other studied steel samples. No additional peak occurred that could be related to the Ti-rich precipitates, probably due to their small volume fraction.

The amount of retained austenite formed after solution treatment and aging at the optimum condition for the present steel was calculated utilizing X-ray diffraction (Equation (1)) and tabulated in Table 5. Table 5 shows that the quantity of retained austenite of the studied steel gradually increased with increasing aging time. The quantity of retained austenite increased due to the coarsening of the intermetallic compounds, which consume the alloying element from the martensite phase. Furthermore, diffusion of the alloying elements from the martensite phase produces martensite with lower alloying elements that decrease the stability of the martensite phase and lead to its transformation to austenite. This can explain the decrease of hardness with increasing aging temperature, as shown in Figure 4. 

### 3.6. EBSD Analysis

The microstructural morphologies observed in the investigated steel corresponded well with the lath-martensite (blue area) structure characterized by EBSD analyses. Figure 8 shows the EBSD IQ-phase map of the experimental sample. The larger blocky reversed γ (red area) formed in martensite blocks are displayed in Figure 8. Moreover, the amount of reversed-austenite was measured to be 13.2 vol.%, while the laves-phase C14 (green area) was measured to be 17 vol.%. The obtained results confirmed the X-ray diffraction test. The black lines indicate the high-angle grain boundary (HAGB) over 15°, which mainly includes the hierarchical boundaries prior-austenite grains, packets, and blocks [47], Figure 8b. Meanwhile, the misorientation below 5° reveals sub-grains colored in red.

Figure 9 presents the EBSD result of the M10AC steel specimen aged at 500 °C for 120 min, revealing the fine austenite (γ) grains were distributed in a ferrite (α-martensite) matrix. The retained γ phases often appeared along the grain boundaries in the α-martensite matrix. Many of the fine-γ grains were oriented along the Z-direction (Figure 9b), forming a {001} texture of the retained γ phase (Figure 9).

The corresponding stereographic projections of Figure 10a depicting the low-index orientations obtained by performing the EBSD analyses are displayed in Figure 10a–d. EBSD analysis revealed the orientation relationship between the fine retained γ grains oriented along the Z-direction and the adjacent α grains in the martensite structure (Figure 10a). As exhibited in the obtained stereographic projections (Figure 10a,b), the fine γ grain had an orientation relation of (ı ı ı)γ//(0 ı ı)α and [ī 0 ı]γ//[ī ī ı]α concerning the adjacent α grain. The stereographic projections (Figure 10c,d) representing the fine γ grain also had a different variant of (ı ı ı)γ//(0 ı ı)α and [ī 0 ı]γ//[ī ī ı]α, concerning another adjacent α grain. The determined orientation relationship corresponds to the Kurdjumov–Sachs (K–S) orientation relationship between lath martensite and austenite and is consistent with the crystallographic features of the lath martensite structure in conventionally quenched maraging steels.

Figure 11 denotes the histogram plot of 10 wt.% Mo AC aged-steel specimen. Furthermore, Figure 11 shows small secondary precipitates of Mo introduced in the final microstructure, with an average diameter of 19 ± 1 nm and narrower size distribution. 

In summary, the experimented steel specimen showed that the Mo-rich phase precipitates were present, with distinctively different sizes. The laves-phase precipitate (MoCr) was observed in the microstructure and found within the martensite laths and along the lath boundaries.

### 3.7. TEM Analysis

From the TEM-image, given in Figure 12 and Figure 13, a high dislocation density was realized in the α-martensite matrix in both solid solution and aged steel specimen. It is clear that the dislocation density decreased while the martensite lath boundaries gradually disappeared with the film-like reversed γ observed at the inter-laths. The above results indicate that the austenite reversion occurred with increased aging time. The precipitate size, on average, was found to be approximately 40 nm in length and 2.5 nm in thickness. The intermetallic compound precipitates, though entirely small, were found to show a variety of fringe contrast that could be seen clearly in the dark field images (Figure 12b) obtained by using precipitate reflections. For specific electron beam directions, these fringes were found to be parallel to the length of the intermetallic compounds precipitates. In other intermetallic compounds, however, the fringes were found to be perpendicular to their intermetallic compound’s length. 

Figure 14 represents the SAED patterns of the intermetallic compound precipitate in this steel sample, which could be indexed in codes of the hexagonal, eta (η) phase-type; η Ni_3_Ti having lattice parameters a = 0.5101 nm and c = 0.8307 nm. Since this phase has been seen to contain Mo, it will be designated as the Ni_3_(Ti, Mo) phase. In addition to the rod-shaped precipitates, the investigated steel sample showed the presence of nearly spherical shape-precipitates, as shown in Figure 12b. Selected area diffraction patterns (SADP) from these precipitated particles could be indexed in the codes of the hexagonal Fe_2_Mo phase (Figure 14).

### 3.8. Effect of Aging Time on Tensile Properties

Figure 15 and Figure 16 illustrate the engineering and true stress-strain curves obtained for Mo-containing maraging steel (10 wt.% Mo), respectively. Figure 15a represents the engineering stress-strain curves for solid solution steel specimens, while, Figure 15b–d represents engineering stress-strain curves for aged steel specimens at 500 °C for 60 min, 120 min, and 240 min, respectively. Figure 15 shows that the steel specimen aged at any aging time shows a remarkable increase in all tensile properties except the ductility (elongation, e_f_ %). The mentioned result, which may be attributed to during aging of the intermetallic compounds, show that coarsening and accumulation led to precipitation of the hardening strengthening process. This strengthening mechanism is diffusion-controlled, depending mainly on the temperature and time. It was noteworthy that the effect of aging time on tensile properties is a concern in this manuscript. Furthermore, the microstructure instability leads to an increase in the emergence rate of several constituents, such as low-alloyed martensite and high-alloyed retained austenite, in addition to the low-alloyed austenite phase. The decrease in the elongation % (e_f_ %) of the tested steel may be due to the high prompt formation, precipitation, and coarsening of intermetallic compounds. Moreover, austenite and martensite in maraging steel have high dislocation density. Precipitation of intermetallic compounds diminishes the dislocation movement, which causes the strengthening of the investigated steel through the particles-dislocations interaction mechanism. The blocking of dislocation movement decreased the ductility (e_f_ %) of the studied steel. On the other hand, non-metallic inclusions may be considered as sites of crack formation and initiation. Therefore, the presence of non-metallic inclusions diminishes the tensile properties of the investigated steel, especially the elongation %, because it has a role as an obstacle of dislocation movement and site of crack initiation and propagation. 

To confirm the previous phenomena, the tensile fractography of the solid solution and the aged M10AC sample (500 °C/120 min) at room temperature is presented in Figure 17. The dimple mode of the fracture on the fracture surface of tensile tested specimens revealed a ductile failure mode. Dimples were prominent in all samples tested. From the fractographic analysis, we concluded that ductile fracture was dominant, and many micro-void formations took place.

It was noteworthy that the changes in both the microstructure and chemical composition of different constituents led to an increase in the transformation potential of the austenitic phase to the martensitic phase. Strengthening may have occurred due to the TRIP (transformation induced plasticity) phenomenon during deformation in the tensile test [46,47]. This assumption was confirmed utilizing the computation of the strain hardening exponent (n-value) for the tested samples. 

Figure 18 reported that all investigated steel showed two stages according to the strain %. The first stage (at low strain %) was for strain accumulation, while the second stage was for phase transformation (induced martensite); additionally, beyond the second stage, instability or necking began. All aged specimens showed a higher strain hardening exponent than solid solution steel specimens at all strain %. The aging process slightly increased the uniform-elongation from 4.31 to 5.47% (>25%) compared to the solid solution process. Due to the precipitation of a high quantity of intermetallic compounds and retained austenite phase. On the other hand, total elongation % (ef %) was reduced from 11.4 to 7.64%. Furthermore, the aging process significantly improved the ultimate tensile strength for all investigated aging times, as shown in Table 6. 

Investigated steels offer additional advantages derived from a unique multiphase microstructure and add retained austenite; in addition, they added intermetallic compounds to the martensite microstructure. The tensile deformation encourages the retained-austenite phase transformation into martensite. The transformation process enhances the strength of the investigated steel, which was explained the effect of the TRIP phenomenon [48,49]. Therefore, this transformation continues with additional deformation as long as sufficient retained austenite is present, allowing the investigated steel to maintain a high average n-value of 0.09235 to 0.1914 over a tensile test, as shown in Figure 18.

The authors suggest that the mentioned characteristic allows for forming more complex geometries and potentially at a reduced thickness to perform the mass reduction. Furthermore, during the deformation of the steel specimen, the retained austenite remaining in the microstructure can subsequently transform into martensite during the event of the collision. This result makes the investigated steels an appropriate choice for parts in crush zones on a vehicle. 

Meanwhile, the n-value for solid solution steel specimen was ~ 0.09235, the n-value for aged steel specimens ranged 0.1601–0.1914, which is comparable to values for TRIP steel (n-value = 0.23) with the same ultimate tensile strength value. It is well-known that the n-value of the material, primarily steel, depends intensely on the different chemistries and technological processing paths used by steelmakers. The n-value is a function of the strain accumulation of the deformation that affects the transformation of the retained austenite. It is well-known that a higher n-value reduces strain gradients, allowing for more complex deformation. Lower n-value concentrates strains, leading to early failure. 

Since different locations in deformed steel pursue several strain tracks with varying amounts of deformation, the n-value for the designed steel could alter with both parts design and location. The modified microstructures of the advanced high-strength steel permit different property relationships to enhance each steel type and grade to particular application requirements.

Figure 19 shows the work hardening rate-true strain diagram. The work hardening rates of the investigated steel at both solid solution and aged condition dropped quickly after yielding, followed by decreasing continuously at higher strains. The solid-solution and aged samples showed similar behavior of work hardening. Figure 19 shows that aged steel samples were characterized by a higher work hardening rate, which is thought to be related to the large fraction of deformed structure. It is speculated that the additional work hardening is affected by the mechanical incompatibility in the plastic strain gradient and heterogeneous structure evolution multiaxial stresses [48,49] during the elastic-plastic transition period. The previous results showed that the investigated steel strengthened due to the austenite transforming into martensite during the deformation process (tensile test). Moreover, the strengthening of investigated steel was explained by the transformation-induced plasticity (TRIP) phenomenon.

The true stress-true strain curves are drawn as in Figure 16 for verifying the instability point, according to the Consideré criterion [50] for necking. It was established that the solid solution steel specimen had low uniform strain (4.31%) and slowly fractured after necking. On the other hand, a considerable elongation (from a true strain of 5.06% to 5.47%) was realized in the aged steel specimens by postponing early localized necking and/or resisting cracking. For more clarity to differentiate between the tested cases, Figure 20 is constructed.

It is noteworthy that the resultant data was limited to Co-free low-nickel maraging steel produced by the double-melting technique vacuum melting followed by electroslag remelting. In addition, the resulting data is limited for this material after forging and heat-treated at the optimum condition demonstrated in the main text of the manuscript. Therefore, to complete the mechanical properties investigation, studying the behavior of the investigated steel at high temperatures needs more attention. Moreover, suggesting the appropriateness of the investigated steel alloy for working as a creep resistance material is the central subject of our further investigations.

## 4. Conclusions

The effect of heat treatment (i.e., solid solution annealing followed by air cooling in addition to aging at variable temperature and time) on tensile properties and microstructure of air-cooled Co-free, low Ni-10 Mo-1.2 Ti maraging steel was investigated. From the presented results, the following conclusions may be summarized.

The low carbon content of the investigated steel leads to very low or no carbide precipitate. High molybdenum contents encouraged lava-phase precipitate with chemical composition formula MoCr and enforced the investigated steel to precipitate η(Ni_3_Ti).The microstructure of aged steel specimens consists of (i) high-alloyed martensite phase, (ii) nickel-rich phase, (iii) low-alloyed retained-austenite phase, (iv) intermetallic compounds, (v) and lavas-phase (MoCr). The presence of this microstructure enhances the tensile properties by precipitation strengthening and TRIP phenomenon.TEM and EBSD studies emphasized that the produced steel has high dislocation density with nano-sized precipitate with an average size ~19 ± 1 nm.EBSD studies revealed that the fine austenite (γ) grains were distributed in a ferrite (α-martensite) matrix. The retained γ phases often appeared along the grain boundaries in the α-martensite matrix. Many of the fine γ grains were oriented along the Z-direction, forming a {001} texture of the retained γ phase.Calculation of the strain hardening exponent (n-value) shows that investigated steel (passing two stages) and its n-value are comparable with the n-value of TRIP steel. Aging the steel specimens at 500 °C and 120 min shows a high work-hardening rate compared with other aged steel specimens.

## Figures and Tables

**Figure 1 materials-15-02136-f001:**
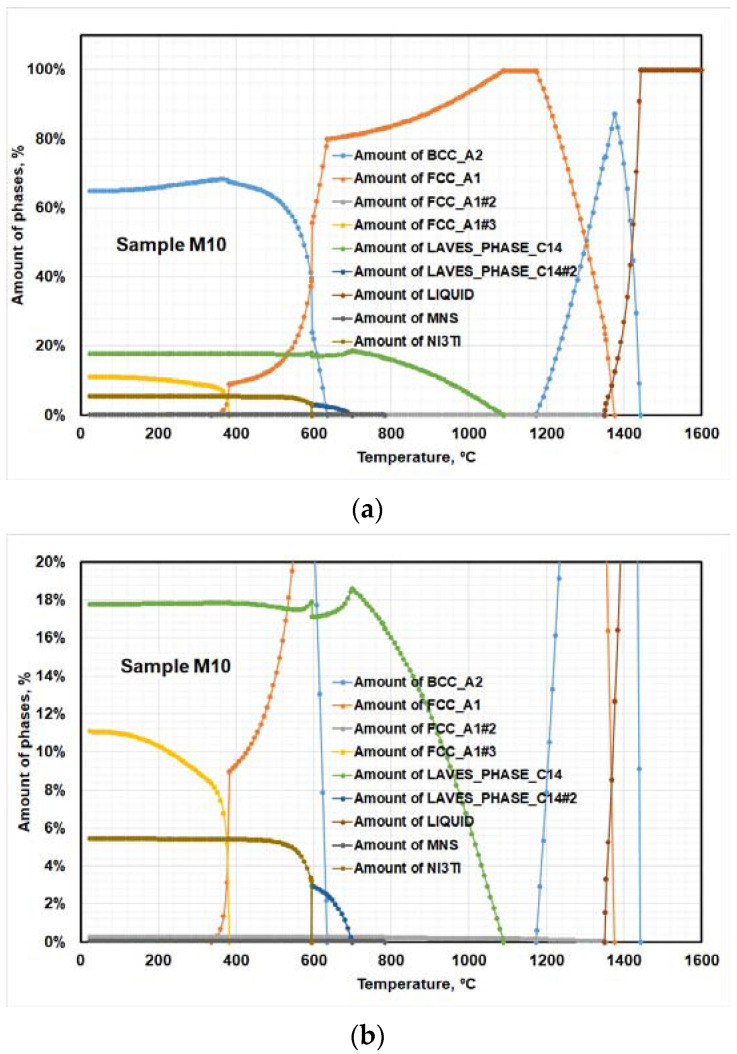
Solidification path of 10 wt.% Mo-containing maraging steel; (**a**) solidification path; (**b**) magnification of (**a**) at a lower amount of phases %.

**Figure 2 materials-15-02136-f002:**
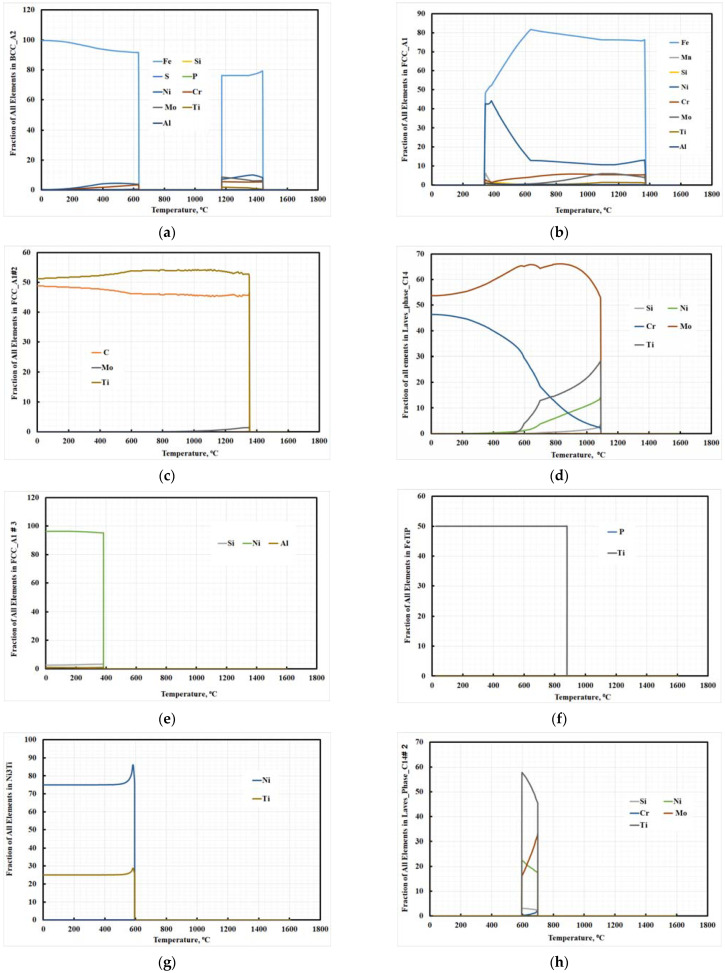
Chemical composition of phases present in investigated steel (**a**) BCC_A2, (**b**) FCC_A1, (**c**) FCC_A2#2, (**d**) Laves_Phase_C14, (**e**) FCC_A2#3, (**f**) FeTiP, (**g**) Ni_3_Ti, (**h**) Lave_Phase_C14# 2.

**Figure 3 materials-15-02136-f003:**
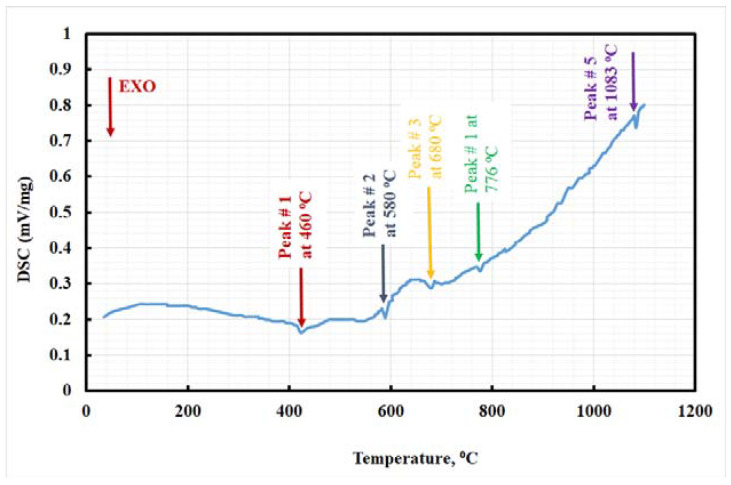
DSC curve of studied 10 wt.% Mo-containing maraging steel.

**Figure 4 materials-15-02136-f004:**
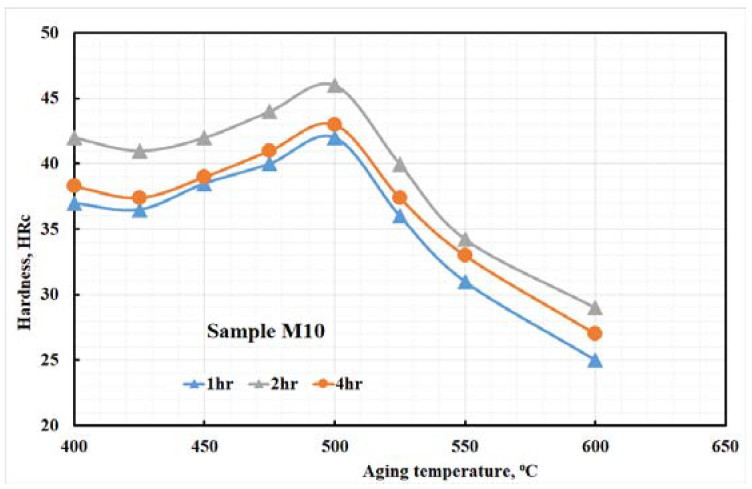
Hardness variation with change aging time and temperature.

**Figure 5 materials-15-02136-f005:**
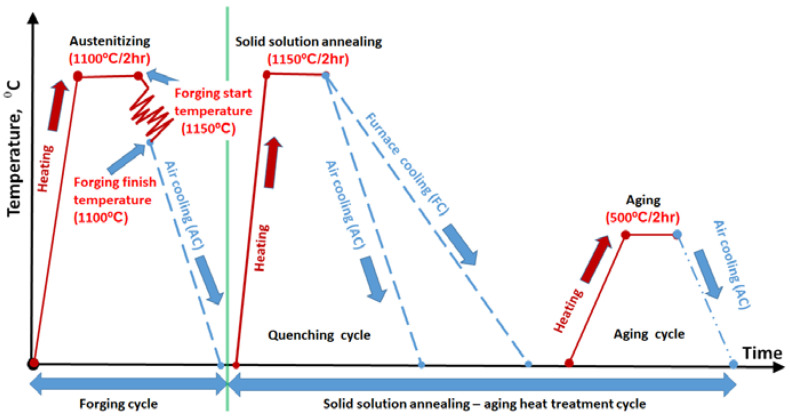
Schematic diagram of forging, solid solution annealing (homogenization), and aging heat treatment cycles for the production of the investigated steel.

**Figure 6 materials-15-02136-f006:**
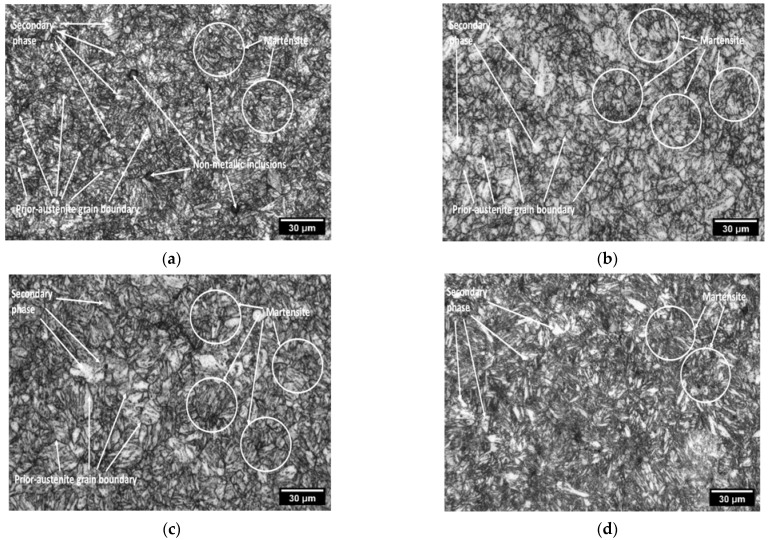
Microstructure of heat-treated sample followed by AC (**a**) solid solution, aged at 500 °C for (**b**) 60 min, (**c**) 120 min, (**d**) 240 min.

**Figure 7 materials-15-02136-f007:**
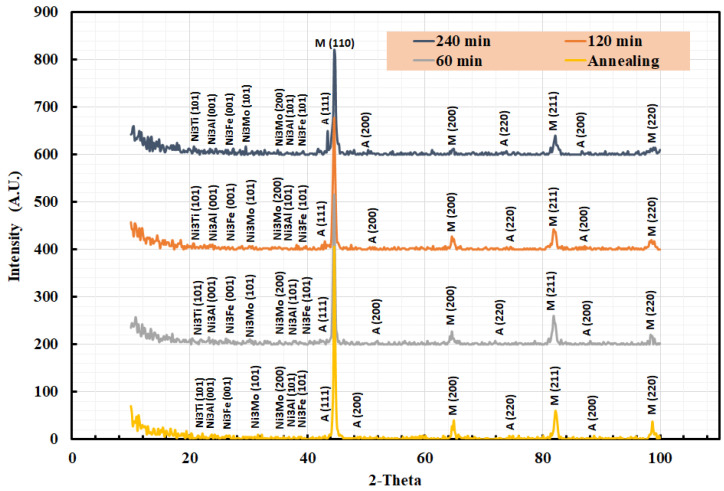
XRD pattern of studied steel air cooling specimen (M10AC) aged at the optimum aging condition.

**Figure 8 materials-15-02136-f008:**
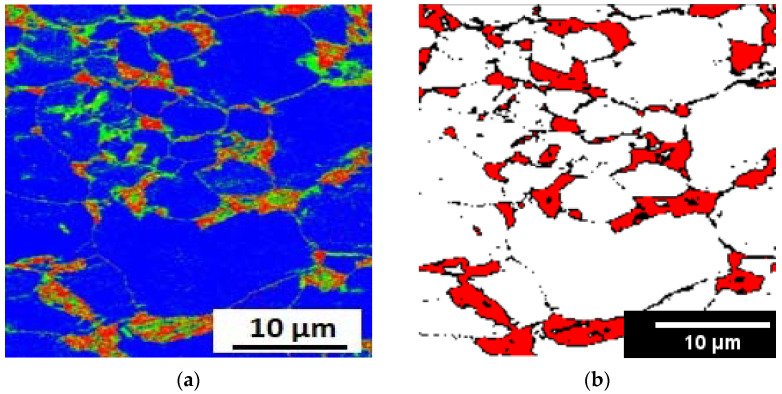
(**a**) EBSD IQ-phase map of M10AC sample aged at 500 °C for 120 min; (**b**) grain boundary classification.

**Figure 9 materials-15-02136-f009:**
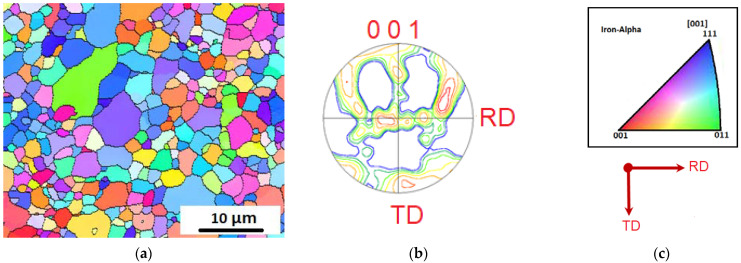
(**a**) Orientation color map of the α and γ phase; (**b**) stereographic projection of the 001 poles in the γ phase. (**c**) key of the subfigures (**a**,**b**).

**Figure 10 materials-15-02136-f010:**
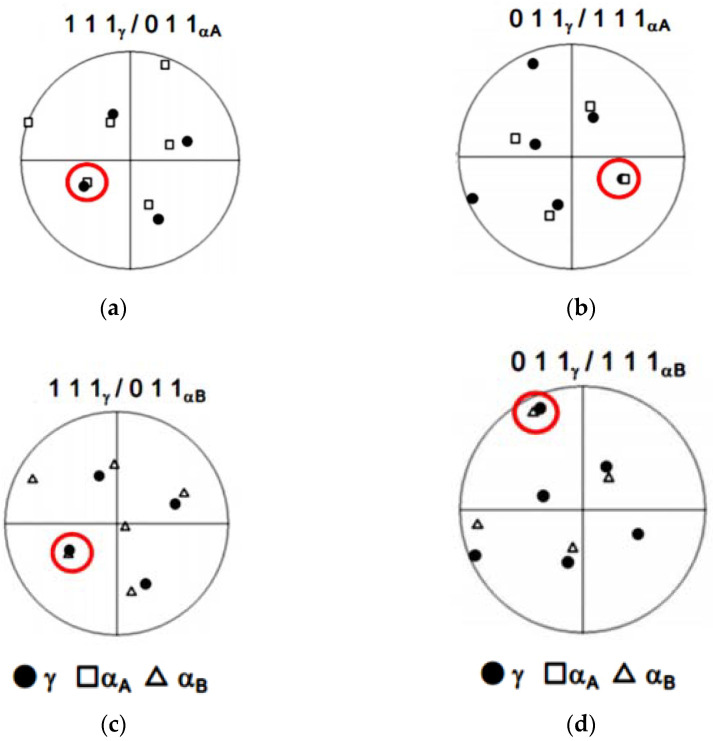
Stereographic projection of the 111_γ_ /011_α_ and 011_γ_ /111_α_ poles.

**Figure 11 materials-15-02136-f011:**
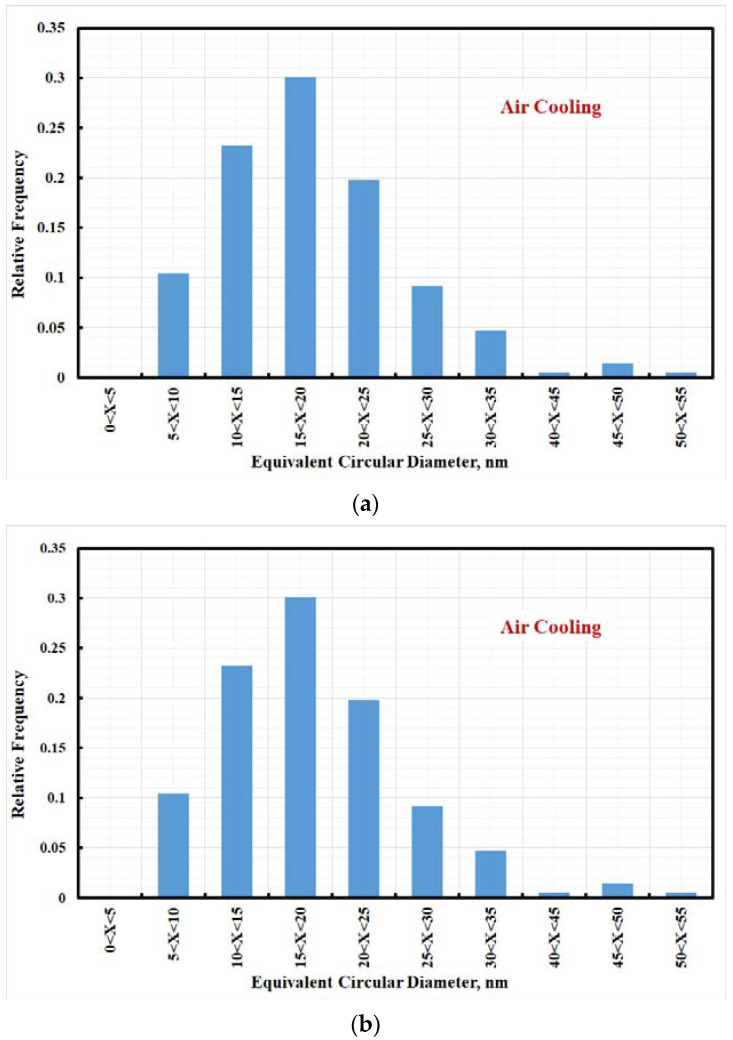
Histogram of air-cooled M10AC steel (**a**) solid solution and (**b**) aged at 500 °C for 120 min.

**Figure 12 materials-15-02136-f012:**
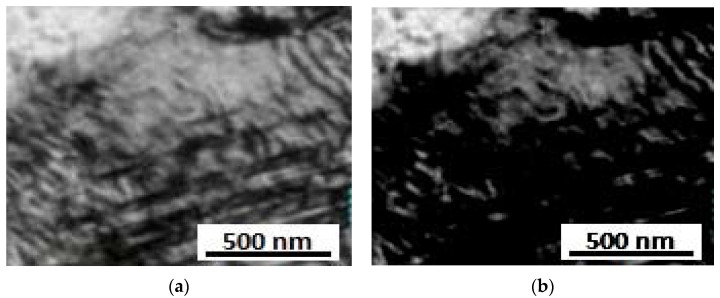
TEM images solid solution maraging steel sample (**a**) bright field, (**b**) dark field.

**Figure 13 materials-15-02136-f013:**
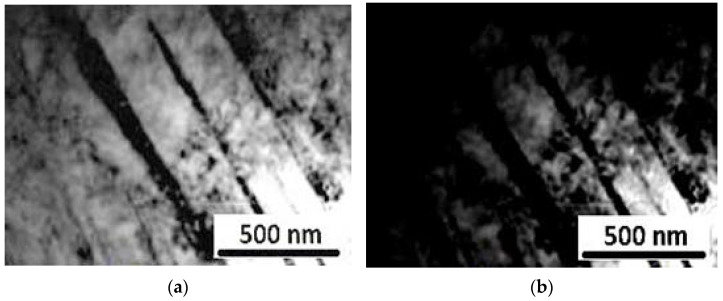
TEM images of aged maraging steel sample (**a**) bright-field, (**b**) dark field.

**Figure 14 materials-15-02136-f014:**
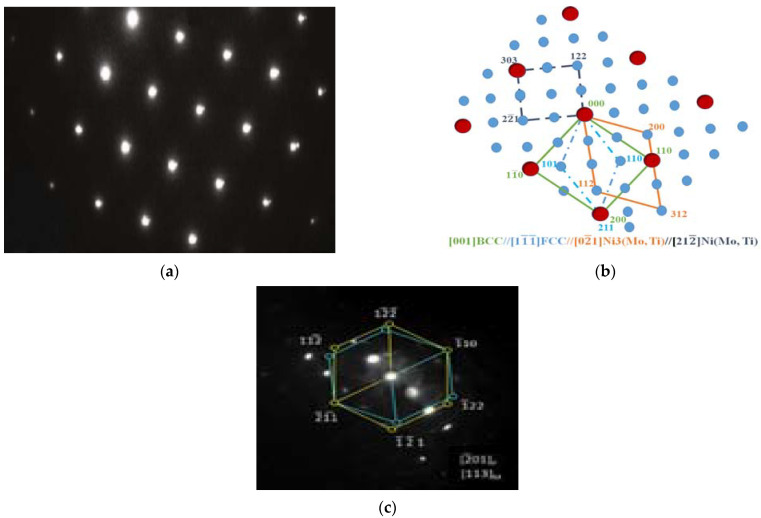
TEM images of maraging steel specimen (**a**) corresponding SAED pattern, (**b**) indexing of diffraction pattern, and (**c**) laves-phase and its index of the diffraction pattern.

**Figure 15 materials-15-02136-f015:**
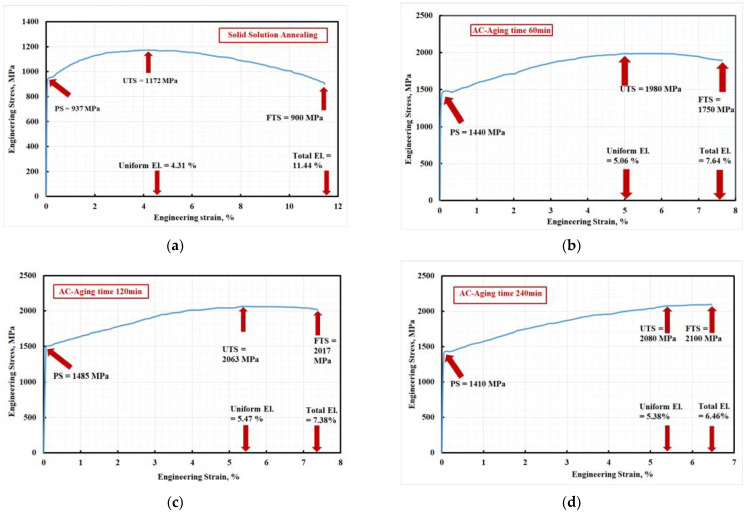
Engineering stress-true strain curves of air cooled 10 wt.% Mo-containing steel: (**a**) solid solution treated and aged at 500 °C for (**b**) 60 min, (**c**) 120 min, and (**d**) 240 min. Where PS = yield strength; UTS = ultimate tensile strength; FTS = fracture tensile strength; El = elongation%.

**Figure 16 materials-15-02136-f016:**
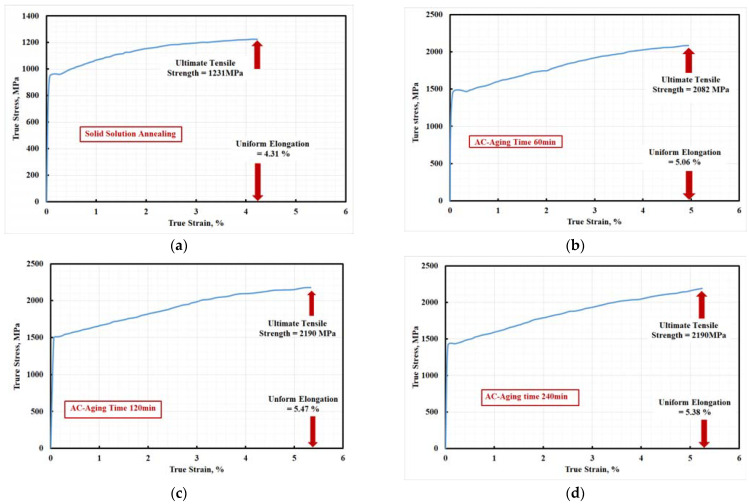
True stress-true strain curves of air-cooled 10 wt.% Mo-containing steel: (**a**) solid solution treated and aged at 500 °C for (**b**) 60 min, (**c**) 120 min, and (**d**) 240 min.

**Figure 17 materials-15-02136-f017:**
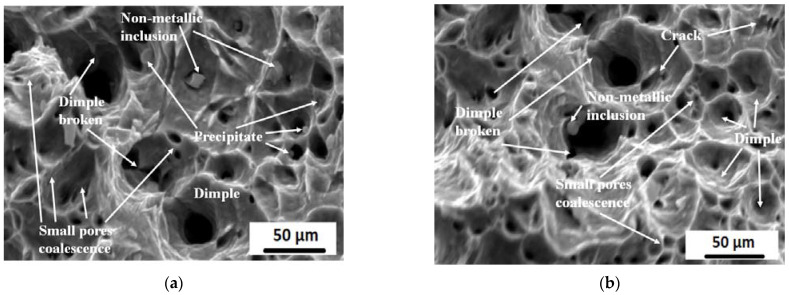
Room temperature tensile fractography of M10 AC sample; (**a**) solid solution; (**b**) aged at 500 °C for 120 min.

**Figure 18 materials-15-02136-f018:**
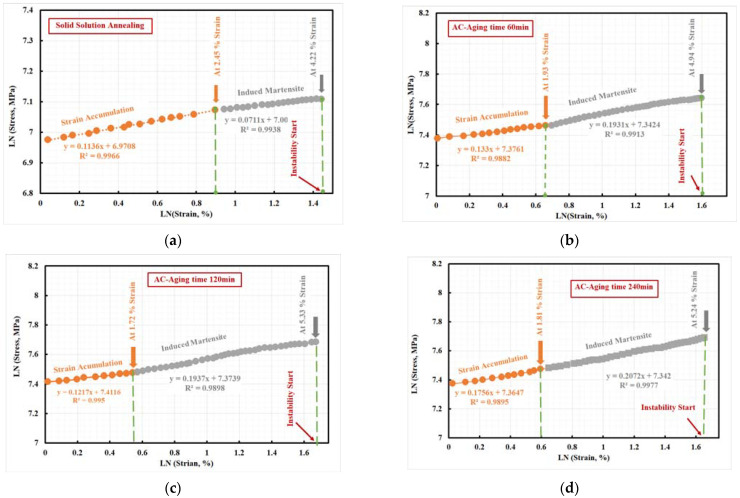
n-value of air-cooled 10 wt.% Mo-containing steel: (**a**) solid solution treated and aged at 500 °C for (**b**) 60 min, (**c**) 120 min, and (**d**) 240 min.

**Figure 19 materials-15-02136-f019:**
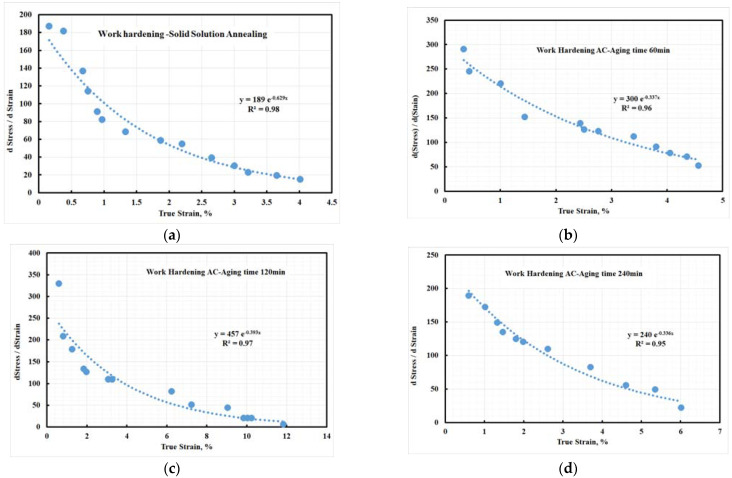
Work hardening coefficient of air-cooled 10 wt.% Mo-containing steel: (**a**) solid solution treated and aged at 500 °C for (**b**) 60 min, (**c**) 120 min, and (**d**) 240 min.

**Figure 20 materials-15-02136-f020:**
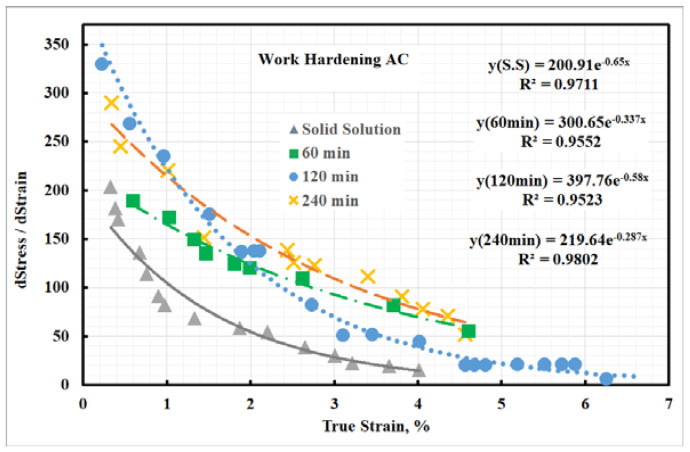
Work hardening rate of investigated steel.

**Table 1 materials-15-02136-t001:** Studied maraging steel chemical composition in wt.%.

	Composition (%)										
Sample	C	Mn	Si	S	P	Ni	Cr	Mo	Ti	Al	Fe
M10AC	0.025	0.12	0.084	0.015	0.009	10.8	4.75	9.8	1.24	0.081	Bal.

**Table 2 materials-15-02136-t002:** Models of the phases considered in the calculations.

Phases	Note	Numbers of Sub-Lattice	Numbers of Sites per Sub-Lattice	Sub-Lattice Species (Va = Vacancy)
Liquid	Liquid	1	1	C, Cr, Fe, Si, Mo, Ni
BCC_A2 (high temperature)	BCC (δ-ferrite)	2	1:3	Fe, Cr, Si, Mo, Va
BCC_A2 (martensite)	BCC (α-ferrite)	2	1:3	Fe, Cr, Si, Mo, Va
FCC_A2 (austenite)	FCC (γ-austenite)	2	1:1	Fe, Mo, Ni, Cr, C, Va
FCC_A2#2	MC (Carbide)	2	1:1	Ti, Mo, V, W:C
FCC_A2#3 (Nickel-rich phase)	FCC (retained austenite)	2	1:1	Ni, Si, Al, Va
Lava-phase_C14	Lava-phase_C14	2	1:3	Mo, Cr, Ni, Va
Lava-phase_C14#2	Lava-phase_C14#2	2	1:3	Mo, Ti, Fe, Si, Cr, Va
Ni_3_Ti	η(Ni_3_Ti)	2	1:3	Ni, Ti, Fe, Mo, Cr, Va
MnS	MnS (non-metallic inclusion)	2	1:1	Mn, Fe, S, Va

**Table 3 materials-15-02136-t003:** Amount of each constituent and the starting phase precipitation temperature, °C of the studied steel.

Constituents	Constituents Starting Temp., °C	Room Temp. Mole Fraction, %
Liquidus	>1442	-
BCC_A2 (high temperature)	1442	-
BCC_A2 (α-martensite)	623	65.382
FCC_A2 (γ-austenite)	1375	-
FCC_A2#2 (MC-carbide)	1354	0.248
FCC_A2#3 (Nickel rich phase- retained austenite)	393	11.15
Laves-phase_C14	1090	17.78
Laves-phase_C14:2	698	-
Ni_3_Ti	596	5.44
MnS	783	0.055

**Table 4 materials-15-02136-t004:** Optimum condition of forging and heat treatment for the production of investigated steel.

Steel No.	Forging		Solid Solution (Homogeneity) Annealing Process		Aging	
	Start Temp., °C	Finish Temp., °C	Temp., °C	Time, h	Temp., °C	Time, h
M10AC	1150	1100	1150	2	500	2

**Table 5 materials-15-02136-t005:** Retained austenite volume fraction, % of studied maraging steel.

Steel No.	Retained Austenite Volume Fraction, %		
	Thermo-Calc	State of Steel	X-ray
M10AC	11	Solid solution, S.S.	10.5
		S.S. + Aging (60 min)	10.9
		S.S. + Aging (120 min)	13
		S.S. + Aging (240 min)	17

**Table 6 materials-15-02136-t006:** Tensile properties of studied maraging steel.

State of the Steel	Time, Min	Tensile Properties, MPa				Strain Hardening Exponent		
		Y.S (MPa)	U.T.S (MPa)	Uniform Strain %	Elongation, e_f_ %	n1-Value	n2-Value	Average n1-Value
Solid Solution	-	937	1231	4.31	11.44	0.1136	0.0711	0.09235
Aging	60	1440	2082	5.06	7.64	0.133	0.1931	0.1601
	120	1485	2175	5.47	7.38	0.1217	0.1937	0.1577
	240	1410	2190	5.38	6.46	0.1756	0.2072	0.1914

## Data Availability

The experimental datasets obtained from this research work and then the analyzed results during the current study are available from the corresponding author on reasonable request.

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
