# Peer review of "Effect of Heat Treatment on Tensile Properties and Microstructure of Co-Free, Low Ni-10 Mo-1.2 Ti Maraging Steel"

_materials, 2022, doi:10.3390/ma15062136_

Round 1

Reviewer 1 Report

  • Add more current references (2020 – 2021).
  • References must be put in Materials journal template. Add the DOI links for all references.
  • Highlight better which is the novelty of the work?
  • Please indicate the location of the manufacturer of all devices in accordance with the journal guidelines. Also add all devices used for testing materials.
  • What is the status of the literature according to your work? Make a comparison between the results obtained by you and another previous research.
  • Why did you not specify where the raw materials were purchased on Materials and Methods?
  • Complete the conclusions with the limitations of the proposed methodology. Also write future research.
  • With parameters from Table 4, make a graphical representation to better observe the treatment diagram.
  • Add Miller indices on Figure 6 XRD pattern.
  • Try increasing the resolution in Figure 16.
  • Generally, the quality of the writing could be improved.

Author Response

Dear reviewer,

Thank you very much for your splendid and precise review on our paper in order to improve its quality. After your comment, we have carefully revised and modified the whole manuscript as per yours’ suggestions and recommendation.

Thank you very much for the very helpful and constructive suggestions and recommendations that really improved our work.

Reviewer 2 Report

Dear Authors.

Please do the following corrections.

Abstract: The abstract should be improved. More consistent, with details the scope of the research and the results obtained. Lines 16 to 21 looks explaining the experimental procedures, which should be added to the experimental section. Please write the abstract more natural manner. 

Line 23: there is a typo error in mentioning the degree Celsius.  

Lines 23 to 24: as you are mentioning the abbreviation for the first time such as YS, UTS. explain them like yield strength (YS), etc.  

"Detailed characterizations were made by conducting tensile tests, hardness tests, SEM, Fractography, XRD, and TEM investigations."---what is the outcome. mention in a few lines.

Introduction parts should be improved. The introduction section is noticed to have a limited literature survey about the research topic. There should be a proper continuation of the literature survey and how the research gap exists on this topic. Similarly, why this research topic is important?. 

Line 104: there is a typo error in mentioning the degree Celsius.  

 There is no graphical representation of experimental setup or samples preparation for sections 2.3/2.4/2.5/2.6. 

Line 230: there is a typo error in mentioning the degree Celsius.  

Line 257: there is a typo error in mentioning the degree Celsius. 

Lines:258. The microstructure in the peak aged conditions consisted of martensite packets within prior-austenite grains. Highlight or point out this in Figure 5 with some lines or circles.  

Figure 5: there is a typo error in mentioning the degree Celsius. 

Line 320: Figure citation is missing.

Conclusions part can be minimized.

Please do the modifications. 

Thank You.

Author Response

Dear reviewer,

Thank you very much for your splendid and precise review of our paper in order to improve its quality. After your comment, we have carefully revised and modified the whole manuscript as per yours’ suggestions and recommendation.

Thank you very much for the very helpful and constructive suggestions and recommendations that really improved our work.

Reviewer 3 Report

This paper focuses on the mechanical properties and microstructure of martensitic steel after aging treatment. Some valuable experimental results are provided, but major modifications are still needed before publication.

  1. Please reconsider whether the “keywords” are suitable. In addition, generally speaking, three to five keywords can be selected, and the number of keywords in the article obviously exceeds the requirements.
  2. There are some obvious misuses of units in this paper. Such as Page 1, line 23 and Page 3, line 104. Please check for similar errors in the full text.
  3. Materials and Methods: it is suggested to combine 2.7 optical observation, 2.8 SEM and 2.10 TEM into one part. They all belong to the content of microstructure characterization. In addition, the tensile and hardness test should also belong to the content of mechanical properties test, which can be put together.
  4. Page 8, lines 223-224. These details have been described in the experiment section and are recommended to be deleted.
  5. 7 Fractography Analysis: there are only a few lines in this section, and the content of fracture can be considered to belong to mechanical properties. Therefore, consider adding the fracture to section 3.6.
  6. Section 3.4 is microstructural analysis by optical observation, but section 3.8 and 3.9 are still microstructural analysis by EBSD and TEM, respectively. Is this an appropriate structure for a paper?
  7. 8 EBSD analysis: (1) Figure 13 shows the EBSD phase map. Here, the author should first point out what color represents what phase. It can also be added to figure 13. In addition, it is suggested to add another grain boundary distribution map in Figure 13, including high angle and low angle grain boundaries, so that the reader can clearly understand the results. (2) Page 10, line 420. “…revealing the fine austenite (γ) grains were distributed in a ferrite (α-martensite) matrix”. Fig. 14 (a) shows an orientation imaging map. Is it better to use EBSD phase diagram to explain? Figure 14 (c) is only the scale of EBSD orientation imaging map, so it is not recommended to be listed separately. (3) Figure 14 (b) provides a pole figure. 10. Which direction does the Z direction correspond to? And X direction? These need to be marked directly on the pole figure, instead of using X and Z.
  8. Conclusion: the conclusion of this paper is divided into 8 points. Appropriate consolidation is recommended. Moreover, it is necessary to make a general summary at the beginning of the conclusion.
  9. Generally speaking, this paper does not form a good system. In the content of microstructure analysis, there is no connection with the previous mechanical properties. Much like a scientific report listing the results of experiments, especially the second half of the article.

Author Response

(The authors gave the same response as above.)

Round 2

Reviewer 1 Report

Significant improvements have been made. The paper can be published.

Author Response

Thank you very much for accepting our Paper.

Reviewer 2 Report

Dear Authors.

Thank you for accepting my comments and modifying them.

The paper can be published as it is.

Thank You.

Author Response

(The authors gave the same response as above.)

Reviewer 3 Report

The manuscript can be accepted now.

Author Response

Thank you very much for accepting our Paper for publication.